# The interplay of DNA repair context with target sequence predictably biases Cas9-generated mutations

Ananth Pallaseni [1,9], Elin Madli Peets [1,9], Gareth Girling[1], Luca Crepaldi[1], Ivan Kuzmin[2], Marilin Moor[2], Núria Muñoz-Subirana[3], Joost Schimmel[3], Özdemirhan Serçin[4], Balca R. Mardin[4,8], Marcel Tijsterman [3,5], Hedi Peterson[2], Michael Kosicki[6,7] ✉ & Leopold Parts [1,2] ✉

Repair of double-stranded breaks generated by CRISPR/Cas9 is highly dependent on the flanking DNA sequence. To learn about interactions between DNA repair and target sequence, we measure frequencies of over 236,000 distinct Cas9-generated mutational outcomes at over 2800 synthetic target sequences in 18 DNA repair deficient mouse embryonic stem cells lines. We classify the outcomes in an unbiased way, finding a specialised role for *Prkdc* (DNA-PKcs protein) and *Polm* in creating 1 bp insertions matching the nucleotide on the protospacer-adjacent motif side of the break, a variable involvement of *Nbn* and *Polq* in the creation of different deletion outcomes, and uni-directional deletions dependent on both end-protection and end-resection. Using our dataset, we build predictive models of the mutagenic outcomes of Cas9 scission that outperform the current standards. This work improves our understanding of DNA repair gene function, and provides avenues for more precise modulation of Cas9-generated mutations.

DNA lesions introduce prompts into the genome that are filled by the repair machinery. Controlling the location of this prompt, and biassing the repair outcomes is the foundation for developing genome editing tools. The versatile CRISPR/Cas9 technology excels at targeting thanks to its RNA guided nuclease activity, and generates double-stranded breaks[1]. These breaks are the most toxic lesions that a cell can experience, necessitating the evolution of a robust repair response[2]. This robustness can come at the cost of accuracy, with mutagenic repair leading to a range of mutagenic outcomes seen at Cas9-induced double-stranded breaks[3], while nuclease-deficient Cas9 technologies can reduce the scope of this damage[4]. The generation of loss-of-function mutations has been used to great effect in basic research on genome function and the more precisely controlled technologies for therapeutic purposes to treat disease[4].

The stochasticity in repair makes Cas9 a somewhat unpredictable tool. There is substantial variety in the repair outcomes observed within and across targeted sites, both in type and size of mutation generated, while the distribution of outcomes is highly reproducible at each target[3,5,6]. It is now well understood how the sequence composition of the target site affects the distribution of outcomes generated by Cas9, and computational tools have been developed to accurately predict both the efficacy of cutting as well as outcome distribution at a given target sequence[7–10]. These tools enable more efficient targeting to create frameshift mutations for knockouts, as well as more precise outcome generation for therapeutic purposes.

[1]Wellcome Sanger Institute, Wellcome Genome Campus, Hinxton, UK. [2]Department of Computer Science, University of Tartu, Tartu, Estonia. [3]Department of Human Genetics, Leiden University Medical Center, Leiden, The Netherlands. [4]BioMed X Institute (GmbH), Heidelberg, Germany. [5]Institute of Biology Leiden, Leiden University, Leiden, The Netherlands. [6]Department of Medicine, University of Cambridge, Cambridge, UK. [7]Lawrence Berkeley National Laboratory, Berkeley, CA, USA. [8]Present address: Research Unit Oncology, Merck Healthcare KGaA, Darmstadt, Germany. [9]These authors contributed equally: Ananth Pallaseni, Elin Madli Peets. ✉e-mail: mkosicki@lbl.gov; leopold.parts@sanger.ac.uk

The editing outcomes can vary across cell types[7], suggesting that there is an avenue for their control that is rooted in repair machinery[11–13]. Three major repair pathways act on a DSB in mammalian cells. Non-homologous end joining (NHEJ) creates small insertions and deletions, microhomology-mediated end joining (MMEJ) exclusively leaves deletions between short stretches of identical sequence ('microhomology'), and homologous repair (HR) perfectly repairs the break with no mutations[2]. The pathways are active at different rates and operate in competition with one another[14,15], providing redundancy in protection. NHEJ and MMEJ are active throughout the cell cycle, and repair the bulk of DSBs, while HR is only active during S phase[16,17]. Their contribution to gene editing has so far been tested using a small number of gRNAs[18–20], and the roles of many involved genes have not been completely elucidated. Observing editing outcomes in multiple sequence contexts in repair deficient backgrounds would advance understanding of DNA repair genes and mechanisms, and options for control.

Here, we systematically measure the impact of repair gene knockouts on Cas9-generated DSB repair outcomes. We analyse mutations created at 2838 target sites in 18 mouse embryonic stem (mES) cell lines, each with a single repair gene knockout. We elucidate how the absence of repair genes modulates Cas9 mutation profiles, associate trends in these profiles with target sequence characteristics, and use this knowledge to build predictive models of repair outcomes for each knockout which outperform existing prediction methods.

## Results

### Measuring Cas9 repair outcomes at scale in knockout cell lines

We measured Cas9-generated mutations at randomly integrated synthetic target sequences within a common sequence context in 18 knockout mES cell lines and three control cell lines (Fig. 1a; Supplementary Table 1). After aggregating data from biological replicates, and filtering for coverage, we compiled an outcome distribution for 2838 target sequences in each cell line (Methods, Fig. 1b, Supplementary Fig. 1). We recovered a total of 132,497,944 reads of mutated sequence across all targets and cell lines (minimum 100 reads per target in every cell line), corresponding to 236,659 distinct mutagenic outcomes (a median of 12 per target per cell line), and calculated the frequency of each outcome in each target, defined as the fraction of mutated reads recovered for that target which match that outcome (and is thus bounded between 0 and 100%). To enable comparisons,

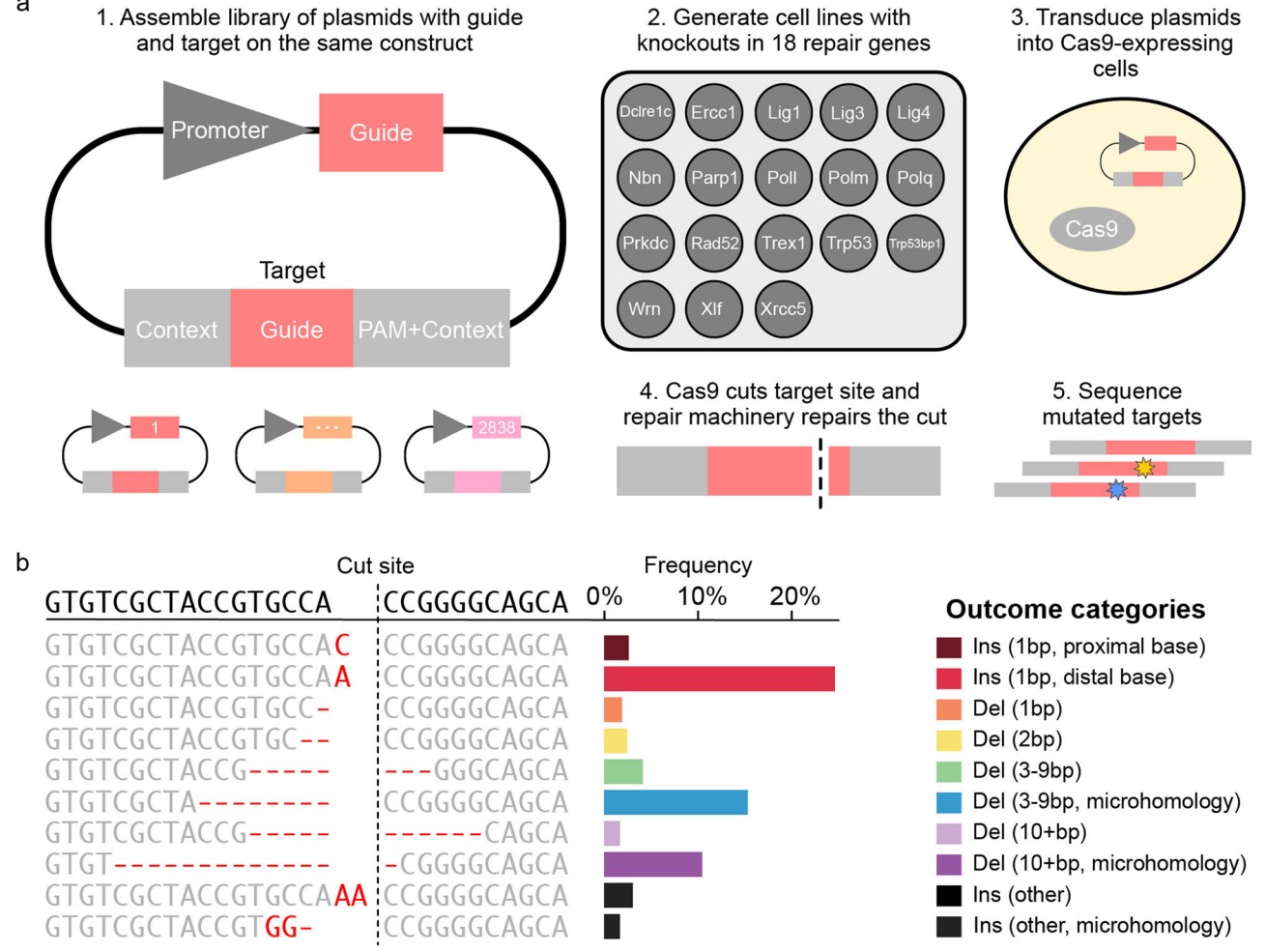

**Fig. 1 | Measuring Cas9 mutational outcomes in DNA repair knockout cell lines at scale. a** A method for high throughput measurement of Cas9-induced repair outcomes. (**1**) Constructs containing both a gRNA and its target sequence (matched colours) in variable context (grey boxes) were cloned into target vectors. (**2**) A panel of Cas9-expressing mouse embryonic stem cell lines deficient in individual repair genes was generated[20] (**3**) Constructs were packaged into lentiviral particles and used to infect the knockout cells. (**4**) Cas9 cuts the target and mutations are created. (**5**) DNA from cells was extracted, the target sequence and context are amplified with common primers, and the mutations in the target are determined by short-read sequencing. **b** An example mutation distribution for a target. The sequence (left, text) and frequency (*x*-axis, bars) of each outcome (*y*-axis). Colours: outcome category. Top sequence: unedited target; vertical dashed line: cut site; red text: altered sequence. NHEJ non-homologous end-joining. MMEJ microhomology-mediated end-joining.

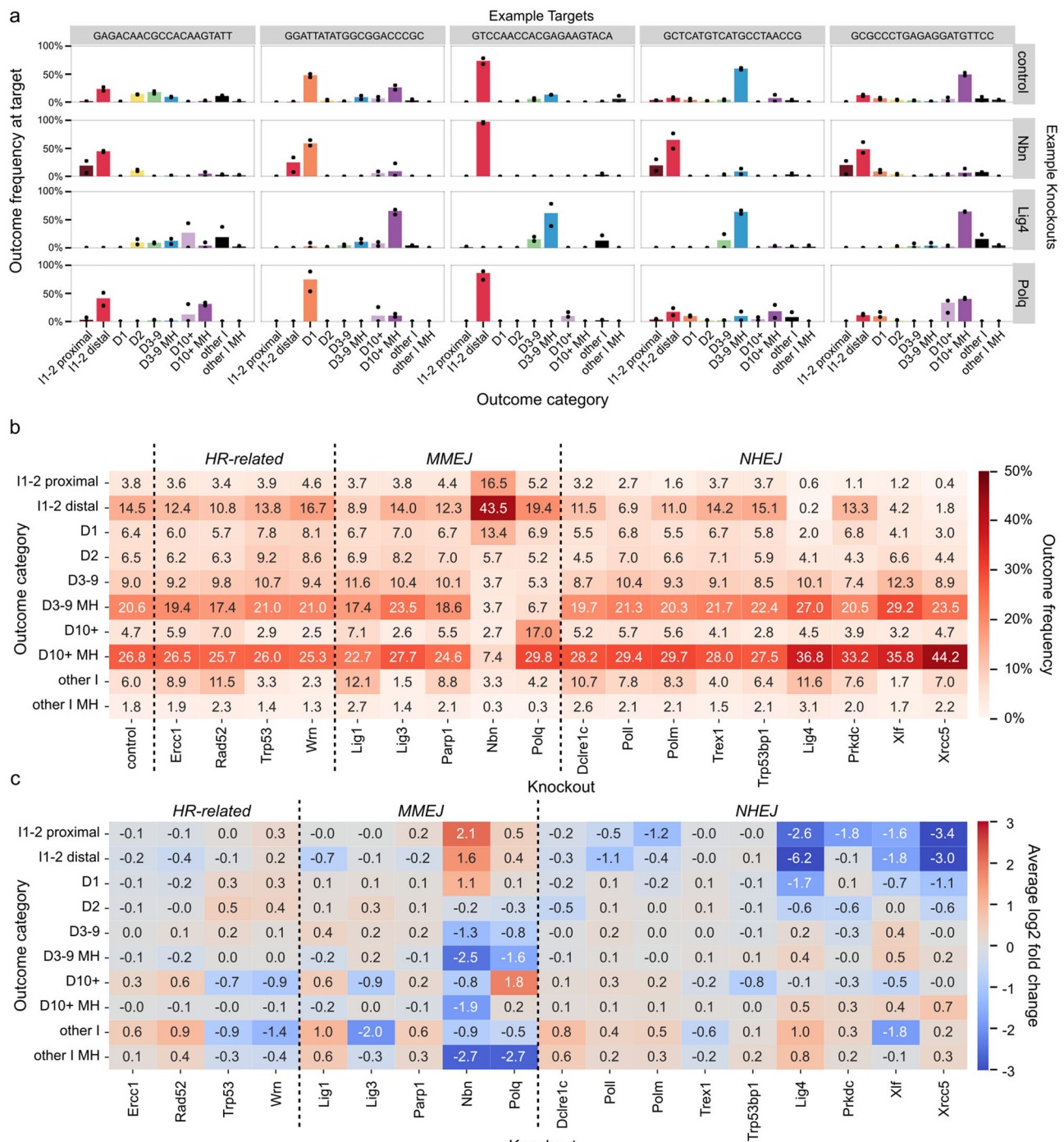

**Fig. 2 | Target outcome profiles are consistently modulated by knockouts.**
**a** The fraction of mutated reads (*y*-axis) of each outcome type (*x*-axis, colours) at various representative targets (columns) in control lines, the *Nbn* knockout, the *Lig4* knockout, and the *Polq* knockout (rows). Dots indicate the fraction of mutated reads in two replicates, which are combined to create the bar. **b** Average fraction of

mutated reads across all targets (annotation, colour) of each outcome category (*y*-axis) observed in each knockout (*x*-axis) organised by repair pathway. **c** Average of log-fold change across all targets (annotation, colour) of each outcome category (*y*-axis) observed in each knockout (*x*-axis) organised by repair pathway. D deletion, I insertion, MH microhomology. Source data are provided as a Source Data file.

we stratified outcomes by type and size into ten groups: 1 bp deletions, 2 bp deletions, medium deletions with or without microhomology (3–9 bp), long deletions with or without microhomology (10+), 1–2 bp insertions at the cut-site matching either PAM-proximal or distal nucleotides, other insertions (incl. insertion-deletions) with or without microhomology (here: match between insertion and flanking target sequence; Fig. 1b).

### Knocking out repair genes modulates outcome profiles
Mutagenic outcomes in control cells were dominated by large deletions with microhomology (10bp+ ; 27%), followed by medium deletions with microhomology (3–9 bp; 21%), 1–2 bp insertions matching PAM-distal nucleotides at the break (15%) and 1–2 bp deletions (13%). We quantified the way repair gene knockouts changed this composition (Fig. 2a, b) by calculating log2 fold change of each outcome

category frequency in each target compared to the control, and taking an average over all 2838 targets (Fig. 2c).

Small insertions and deletions are a hallmark of mutagenic repair of DSBs resulting from NHEJ repair[21]. Consistent with this expectation, we found that knockouts of core NHEJ genes *Lig4*, *Xrcc5* or *Xlf* led to a marked decrease in 1–2 bp insertions and 1 bp deletion, with concomitant increase in medium and large deletions (3bp+). For example, frequency of 1–2 bp insertions and 1 bp deletion was down from 25% in controls to 9.5% in *Xlf* knockout, while medium and large deletions increased from 61% in control to 81% in *Xlf* knockout (Fig. 2b). Core NHEJ knockouts differed in their impact on specific outcome categories. For example, *Xrcc5* knockout resulted in a higher frequency of large deletions (10bp+; 44%) than *Xlf* or *Lig4* knockouts (36–37%), while *Lig4* knockout depleted 1–2 bp insertions matching PAM-distal nucleotides more comprehensively than the other two knockouts (0.2% remaining vs 4.2% in *Xlf* and 1.8% in *Xrcc5* knockout). Knockouts of other NHEJ genes, such as *Prkdc*, *Poll* and *Polm*, had consistent, but more specific and overall milder effects, which we discuss in more detail in later sections.

*Polq* is a core MMEJ gene that both creates homologies through polymerase activity of its gene product and enforces their usage during repair. Consistent with that role and previous observations[22], *Polq* knockout strongly decreased the frequency of medium deletions with microhomologies compared to control cells (3–9 bp; 21% to 7%, Fig. 2b), while increasing occurrence of non-homologous large deletions (10bp+; 5% to 17%). Another gene essential to creating MMEJ outcomes is *Nbn*, whose product leads to medium and long resection as a part of MRN complex[23,24]. Its knockout produced the strongest effect of all tested genes in the our panel, suppressing medium and large deletions (3bp+; 61% to 18%) and resulting in profiles enriched in 1–2 bp insertions and 1 bp deletion (25% to 73%). We investigate the relationship between deletion sizes and microhomology usage in the context of *Nbn* and *Polq* knockouts in more depth further below.

The other MMEJ-associated genes in our panel (*Lig1*, *Lig3* and *Parp1*) and other repair genes (*Dclre1c*, *Wrn*, *Trex1*, *Trp53*, *Trp53bp1*, *Rad52* and *Ercc1*) did not substantially affect the major outcome categories (Fig. 2b), but had an effect on large non-homologous deletions (10bp+) and the non-homologus insertion-deletions ('other I'). However, both of these categories involved outcomes that were collectively (5–6%) and individually infrequent in control cells, with a mean frequency of 1% per outcome, compared to 5–10% for each of the other deletion categories (including 1–2 bp deletions). We therefore speculate that the different frequencies of these outcomes in knockout lines more likely represent stochastic variation in rate of rarer events due to lower sequencing depth of some cell lines, rather than effects of biological interest.

## Variable response to knockouts distinguishes mutation classes

To explore how DNA sequence determines DSB repair results, we used outcome frequency changes in the 18 knockouts for grouping similar events. To do so, we first removed outcomes that were not observed in the control, or in the majority of the knockouts to avoid noise associated with stochastic dropout of low frequency events. The removed outcomes were relatively rare (mean frequency of 2.5% per outcome, total frequency of 25% per target) and belonged primarily to collectively rare categories such as large non-homologous deletions (10bp+) and complex insertion-deletion outcomes ('other I' and 'other I MH'). The log2 fold changes of remaining outcome categories in response to knockouts remained representative (Supplementary Fig. 2). We then embedded the log2 fold changes to 18 gene knockouts of the remaining 18,105 unique outcomes in two dimensions using Universal Manifold Approximation and Projection (UMAP; median 6 outcomes per target; Fig. 3a). Distance between two outcomes in the UMAP

representation reflects the similarity of log-fold change in their frequencies across knockout lines, indicating a shared response to DNA repair deficiencies. Outcomes of the same category, similar frequency or similar size tended to co-localise within the embedding (Fig. 3a–c). The first UMAP component broadly separated NHEJ-sensitive 1–2 bp indels from MMEJ-sensitive medium and large deletions (3 bp+), and could thus be loosely interpreted as NHEJ-MMEJ axis (Fig. 3b). The second one correlated with outcome frequency in the control cells (Fig. 3c).

Using the UMAP embedding, we grouped the outcomes into seven clusters (Fig. 3d). Most of the clusters were primarily composed of one or two closely related outcome categories and were very strongly depleted by one or two knockouts (Fig. 3e–g; Supplementary Fig. 3). For example, cluster 3 was composed primarily of 1–2 bp indels and was strongly depleted in absence of NHEJ components such as *Lig4* and *Xrcc5*. However, clusters and outcome categories did not always coincide. Outcomes from the same category were sometimes split between clusters, implying divergent regulation, e.g., 1 bp insertions matching PAM-distal nucleotides in clusters 3 and 4 (Fig. 3a). We explore the details of the clusters in the following sections to tease out the DNA sequence dependencies of DSB repair.

## Insertions matching PAM-distal base at the CRISPR/Cas9 break are dependent on *Polm* and *Prkdc*

Single base insertions made up a considerable fraction of outcomes in control cells (17%). We investigated in detail which factors influence their prevalence, nucleotide identity and response to DNA repair gene knockouts.

First, we split the 1 bp insertions in control cells based on their match to nucleotides flanking the cutsite (Fig. 4a). We found a strong correlation between flanking nucleotides and (1) the nucleotide identity of the 1 bp insertion, (2) the total fraction of 1 bp insertions at a given target and (3) the relative amount of 1 bp insertions matching the PAM-distal nucleotide. Nearly 70% of 1 bp insertions matched the PAM-distal nucleotide only ('PAM-distal insertions'), while around 13% matched the PAM-proximal nucleotide only ('PAM-proximal insertions'), compared to random expectation of around 19% for each of these outcomes. Targets with a PAM-distal thymidine or adenosine (T or A) were almost three times as likely to result in a 1 bp insertion as ones with a PAM-distal guanidine or cytidine (G or C; 29% vs 9.7%; Fig. 4b). The fraction of PAM-proximal insertions also differed between targets flanked by different nucleotides (Fig. 4c). In general, 1 bp insertions were biased towards matching a flanking T and away from matching a flanking G. Consistent with this, targets flanked with these two nucleotides—T/G and G/T—showed most extreme biases, with 0.5% and 56% PAM-proximal insertions, respectively. We conclude that 1 bp insertions were strongly biased towards matching the PAM-distal nucleotide and that both total frequency of 1 bp insertions and relative frequency of PAM-proximal insertions in control cells were strongly influenced by the nucleotides at the cutsite.

Next, we investigated the role of knockouts. The 1 bp insertions were depleted in Xrcc5 and *Lig4* knockouts (down to 2% and 0.5%, respectively from 17% in controls), reduced in other NHEJ knockouts (*Polm*, *Poll*, *Xlf*; down to 5–9%) and enriched in *Polq* and *Nbn* knockouts (up to 24% and 55%, respectively). Interestingly, some knockouts also had a differential impact on PAM-distal and PAM-proximal insertions (Fig. 4d). In particular, *Prkdc*, *Xrcc5* and *Polm* deficient cells had substantially fewer PAM-proximal insertions (<4.7%) than controls (9.4%). *Polm* and *Prkdc* knockouts specifically depleted PAM-proximal insertions (LFC of −1.5 and −2.3, respectively; Fig. 4e), with only a minor impact on PAM-distal ones (LFC of −0.3), while *Xrcc5* depleted both types, with a stronger impact on PAM-proximal insertions (LFC = −4.2 for proximal vs −3.1 for distal; Fig. 4e). *Lig1*, *Papr1*, *Nbn* and *Xlf*

 

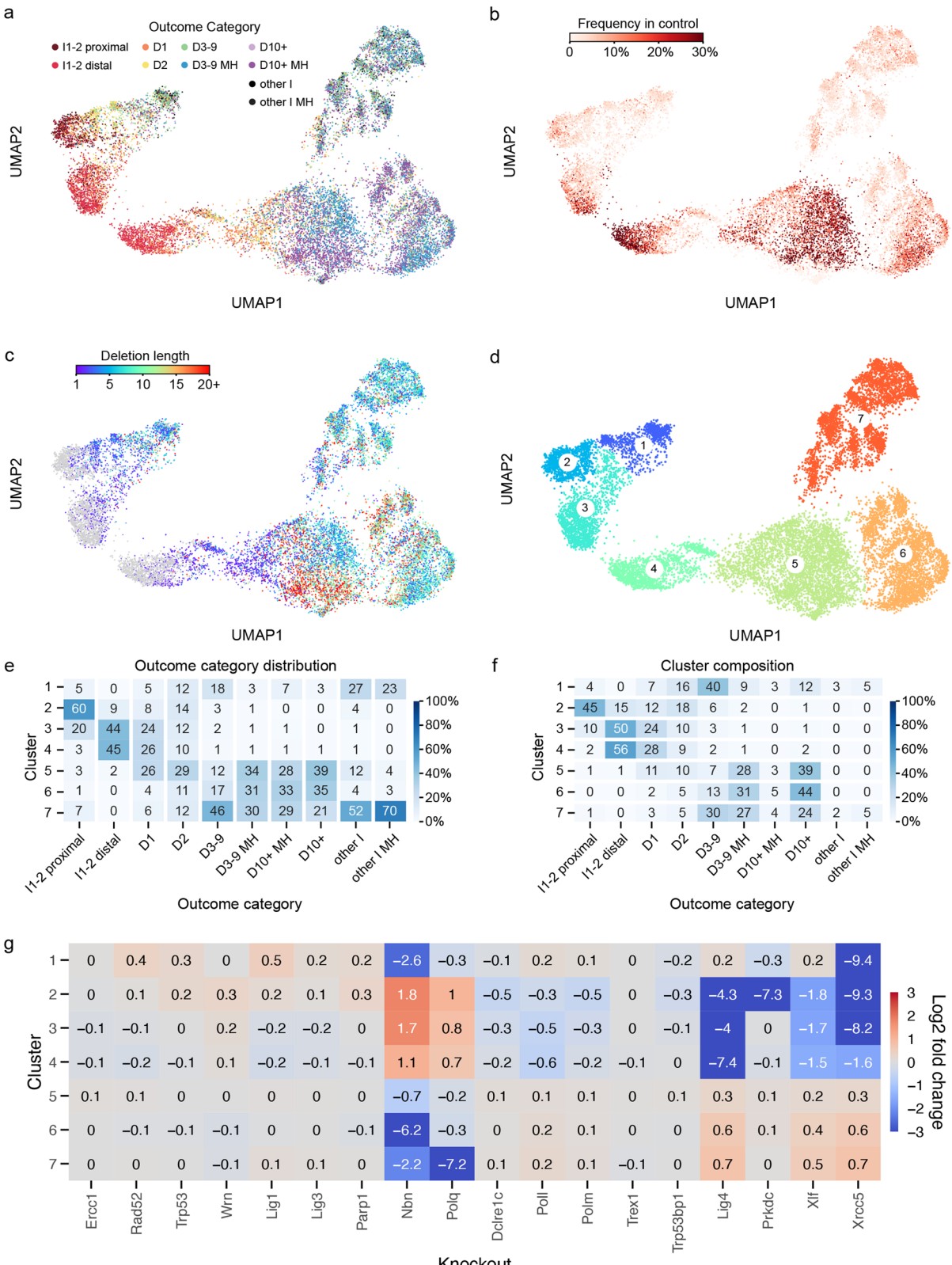

**Fig. 3 | Unbiased analysis of mutagenic outcomes using UMAP projection.**
UMAP embedding of repair outcomes were coloured by: (**a**) outcome category, (**b**) number of deleted basepairs in the outcome (grey indicates no deletion), (**c**) frequency of outcomes in control cells or (**d**) cluster assignment. **e** Distribution of outcome categories among clusters (columns add to 100%). **f** Composition of each cluster in terms of outcome categories (rows add to 100%). **g** Enrichment or depletion of outcomes within clusters across knockouts, normalised across all outcomes within each knockout. D deletion, I insertion, MH microhomology. Source data are provided as a Source Data file.

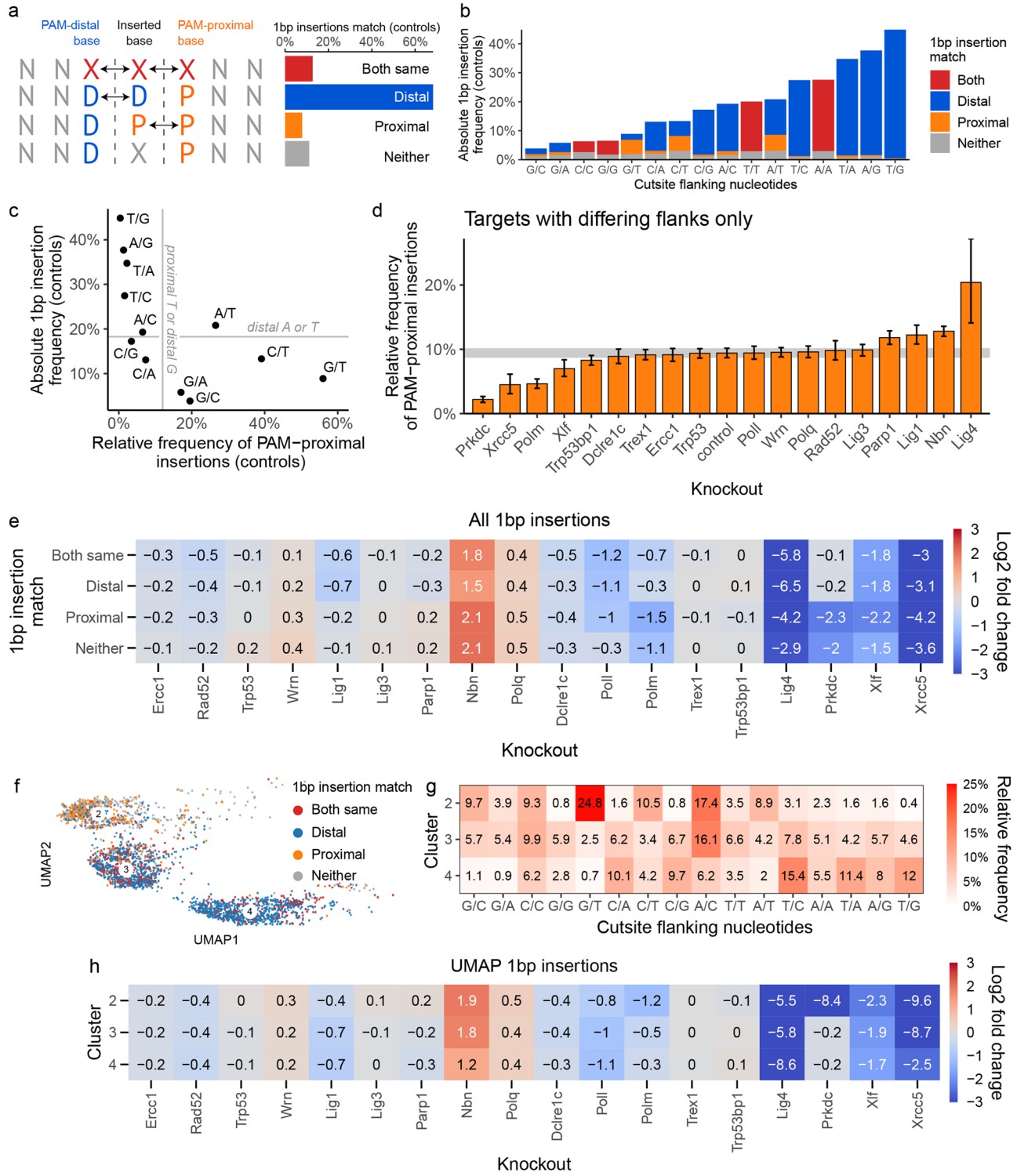

**Fig. 4 | Single basepair insertion outcomes. a** Most 1 bp insertions at the cut site matched the PAM-distal nucleotide in control cells. Dashed line represents Cas9 cutsite between 4th and 3rd basepair upstreams of the PAM sequence. First row represents the case in which basepairs flanking the cutsite are identical to each other, and to the inserted base. **b** Absolute frequency of 1 bp insertions in control cells, by cutsite flanking nucleotides and match between inserted nucleotide and flanking nucleotides. **c** Relationship between absolute 1 bp insertion frequency in control cells and relative frequency of PAM-proximal insertions. Only targets with differing flanking nucleotides included. **d** Relative frequency of PAM-proximal 1 bp insertion across knockouts. Bar height is mean, error bars are 95% bootstrap confidence intervals (*N* = 1000 bootstraping repeats). Grey horizontal band is 95% confidence interval of control cells. **e** Enrichment or depletion of 1 bp insertions across knockouts, segregated by match between insertion and flanking nucleotides. Frequency of 1 bp insertions matching the PAM-proximal nucleotide was diminished in the absence of *Prkdc* and *Polm* compared to controls, while PAM-distal insertions were not affected by these knockouts. **f** Single basepair insertions coloured by the match between inserted nucleotide and the cutsite flanking nucleotides, in UMAP projection. Only clusters 2, 3 and 4 are shown (90% of 1 bp insertion outcomes). **g** Relative frequency of targets within clusters, segregated by cutsite flanking nucleotides, in control cells. Only targets whose 1 bp insertion outcomes are >85% contained within a given target are included. **h** Enrichment or depletion of 1 bp insertions within clusters 2, 3 and 4 across knockouts. Source data are provided as a Source Data file.

knockouts had more modest effects. *Lig4* deficiency nearly completely depleted 1 bp insertions (0.5%), but also resulted in a strong skew towards PAM-proximal ones (20%). *Polq* and *Poll* knockouts affected the total frequency of insertions, but had no significant effect on the balance between distal and proximal bases. We conclude that knockouts can have a strong impact on both the overall frequency and PAM-proximal skew of 1 bp insertions, and that *Polm* and *Prkdc* specifically modulate the latter.

Finally, we examined 1 bp insertion outcomes in the UMAP clustering. More than 90% of 1 bp insertions were in clusters 2, 3 and 4, and these clusters were >50% composed of 1 bp insertions. PAM-proximal insertions were predominantly found in UMAP cluster 2 (60%), with a N/T target bias consistent with this class of insertions and specific sensitivity to *Polm* and *Prkdc* knockouts also previously observed to affect these outcomes (Fig. 4f–h). PAM-distal insertions were split between clusters 3 and 4 (approximately 45% in each; Fig. 4f), implying they could be generated by different repair processes. Indeed, targets in cluster 3 had a different sequence bias than those in cluster 4 (more G/N and A/C; Fig. 4g), and were nearly completely dependent on *Xrcc5* (LFC = −8.7 vs LFC = −2.5, Fig. 4h).

We conclude that 1 bp insertions are likely generated by three repair processes—specifically *Prkdc-Polm* dependent one for PAM-proximal insertions, strongly *Xrcc5*-dependent one for some PAM-distal insertions with specific sequence biases, and partially *Xrcc5*-dependent one for remaining PAM-distal insertions.

## Deletions between large microhomologies are dependent on *Nbn* but not *Polq*

The majority of outcomes in control cells in this screen (61%) were medium and large deletions (3bp+), most of which were associated with microhomology (77%). These outcomes were primarily affected by *Nbn* and *Polq* knockouts. *Nbn* depletion resulted in reduction of medium and large deletions and an increase in 1 bp insertions and 1 bp deletions, while *Polq* knockout mostly depleted medium deletions with moderate amounts of microhomology and led to creation of large deletions (10bp+) with little or no microhomology (Fig. 5a). This is consistent with NBN (*Nbn* product) enabling resection as part of the MRN complex[25] and Polθ (*Polq* product) limiting the size of resulting deletions through use of microhomology.

To understand the relationship between deletions and microhomology better, we modelled the frequency of deletions as a function of microhomology length and distance between the ends of the microhomologous sequences (Methods; Fig. 5b). Deletions utilising longer microhomologies that were closer to each other, were more likely to occur than those using shorter, more distant microhomologies[7]. These trends varied little across knockouts, with the exception of lower baseline frequency in *Nbn* knockout for all sizes of microhomology, and in *Polq* knockout for short microhomologies (Fig. 5b, Supplementary Fig. 4). We speculate that the increased affinity of longer microhomologies lends itself more readily to direct annealing of the broken ends after resection and needs less assistance from Polθ, which is mostly required for shorter stretches.

To understand repair processes leading to medium and large deletions we examined their properties in UMAP clusters 5, 6 and 7, in which >97% of them were found (Fig. 3a, c, e). Cluster 5 was the largest (45% outcomes in control cells) and most heterogenous one of all clusters, containing nearly exclusively microhomologous deletions of all sizes, including 1–2 bp deletions (Fig. 5c, d). Nearly all (92%) large deletions (10bp+) with extensive microhomology (10bp+) in the screen were found in this cluster, and made up a large fraction of its outcomes (27%; Supplementary Fig. 5). While all other clusters were depleted at least 30-fold (LFC < −5) by some genetic perturbations, the strongest effect on cluster 5 was of LFC −0.7 due to *Nbn* deficiency (Figs. 3f, 5e). This relatively low average depletion rate was not merely an artefact of cluster heterogeneity, since the two largest outcome groups that made up cluster 5, medium and large deletions with microhomology, were also only moderately depleted, when considered separately (LFC of −1.4 and −1; Supplementary Fig. 6). Medium, but not large deletions with microhomology within this cluster were also moderately sensitive to *Polq* deficiency (LFC = −0.8 vs LFC = 0; Supplementary Fig. 6). The large deletions in cluster 5 may therefore arise from the Polθ-independent, but MRN-dependent HR pathway[26].

Clusters 6 and 7 were both primarily composed of medium to large deletions (3bp+; Fig. 5c, d). However, cluster 7 contained less frequent outcomes and had smaller deletions with less microhomology (50% of cluster were non-microhomologous events, Fig. 3f), was less sensitive to *Nbn* knockout (LFC = −2.2 vs LFC = −6.2 for cluster 6), but strongly dependent on *Polq* (LFC = −7.2 vs LFC = −0.3; Fig. 5e). This is consistent with Polθ (product of Polq) enforcing closure of resected DSB ends using microhomology, preventing further loss of genetic information due to resection.

We conclude that the majority of deletions initiated by NBN-resection in this screen fall into one of three groups—rare medium to large deletions with resection limited by Polθ (cluster 7), more frequent medium to large deletions whose ends were joined with no or very limited Polθ involvement (cluster 6) and a range of small to very large deletions that end in annealing of extensive microhomologies (cluster 5), likewise with limited Polθ requirement.

## A subset of non-homologous deletions is dependent on both *Xrcc5* and *Nbn*

The majority of UMAP clusters could be interpreted as representing some aspect of NHEJ or MMEJ repair. Cluster 1 was an exception, being depleted in both *Nbn* and *Xrcc5* knockouts, which represent the mutually exclusive results of end-resection leading to MMEJ and end-protection leading to NHEJ (Fig. 3g). We confirmed that this was not an artefact of averaging over outcomes with different dependencies, as most of the outcome categories within this cluster were *Xrcc5* and *Nbn*-dependent (Supplementary Fig. 6). The dominant outcomes in cluster 1 were small and medium sized (1–9 bp) deletions with limited or no microhomology (0–1 bp; Fig. 6a). One possible explanation for the paradoxical dependency on both *Xrcc5* and *Nbn* could be unidirectional resection, with *Xrcc5* protecting one end of the break and *Nbn* resecting the other. Indeed, we found that the directionality of resection in non-homologous deletions in cluster 1 was higher than average for the library as well as in comparison to cluster 7, which also harboured a similar spectrum of nonhomologous deletions (Fig. 6b). Therefore, we speculate that DSBs in cluster 1 were a result of unidirectional resection.

## Outcome profiles are predictable in knockout contexts

Having quantified the sequence determinants of Cas9 outcomes in repair-deficient contexts, we set out to build computational predictors of their behaviour. We used the FORECasT model[7], a multiclass regression that predicts the frequency of outcomes at a target ('outcome profile') from sequence features. We split the data into training and test sets, and trained one FORECasT model per knockout using the training data. Kullback-Leibler (KL) divergence between the predicted and the measured outcome profile was used as the loss function (Methods). The distribution of KL divergences from predictions was close to the one observed between replicates (average divergence 1.53 vs 1.25 between replicates; Fig. 7a). For predictions of outcome profiles in knockout cells, the models trained on relevant knockout data outperformed the model trained on control cells and the original FORECasT model. The difference was particularly strong for knockouts with strong phenotypes, for example the *Nbn* knockout (average divergence 1.21 vs 3.18 for control line and 3.20 for FORECasT; Fig. 7a).

We quantified our prediction performance on held out test data. Pearson's correlation between predicted and observed outcome

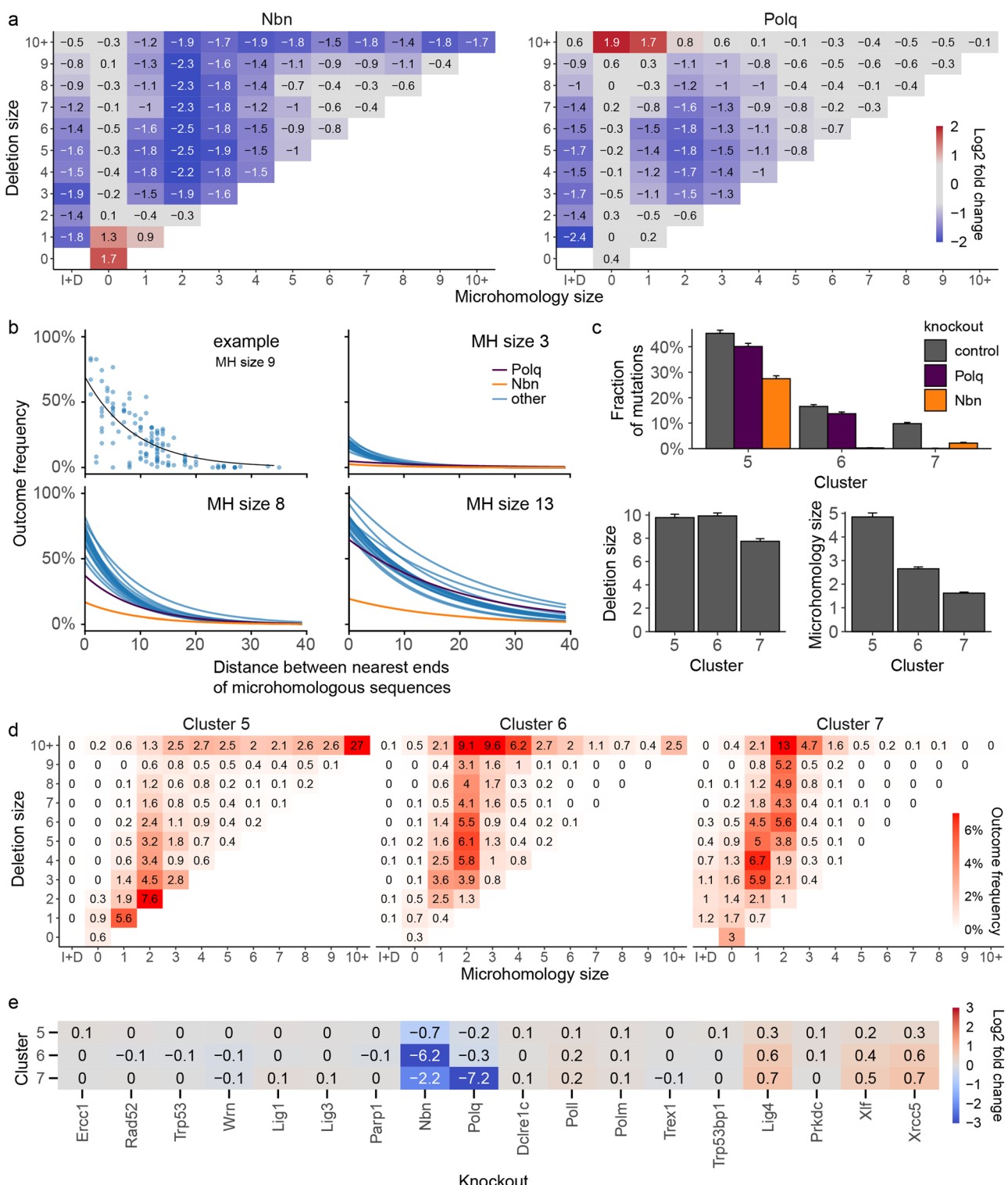

**Fig. 5 | *Nbn* and *Polq* deletion outcomes. a** Relative enrichment or depletion of different groups of outcomes in *Polq* and *Nbn* knockouts, relative to controls. I +D = outcomes that contain both insertions and deletions. Box with 0 bp deletion size and 0 bp microhomology = pure insertions. A pseudocount of 0.1% was added to all cells prior to calculating the log2 fold change to reduce variability. **b** The relationship between outcome frequency, size of microhomology (MH) sequence, and distance between microhomology sequences is different in *Nbn* knockouts. Frequency of microhomology deletion (*y*-axis) vs the distance between micro-homology sequence (*x*-axis) for a single length of microhomology (panels) for each knockout (blue lines), with *Nbn* (orange) and *Polq* (purple) highlighted. First panel

shows the data points used to fit regression lines for a single knockout and microhomology size. **c** Average frequency (top), microhomology size (left) and deletion size (right) of outcomes in clusters 5–7. Bars are 95% confidence intervals (2 × standard error of the mean, *N* = 2838 targets). **d** Composition of clusters 5–7 in terms of microhomology (*x*-axis) and deletion size (*y*-axis). Frequencies in each cluster add up to 100%. See (**a**) for additional definitions. See Supplementary Fig. 6 for remaining clusters and across-cluster quantification. **e** Enrichment or depletion of outcomes in knockouts, broken by cluster. Fragment of Fig. 3g. Source data are provided as a Source Data file.

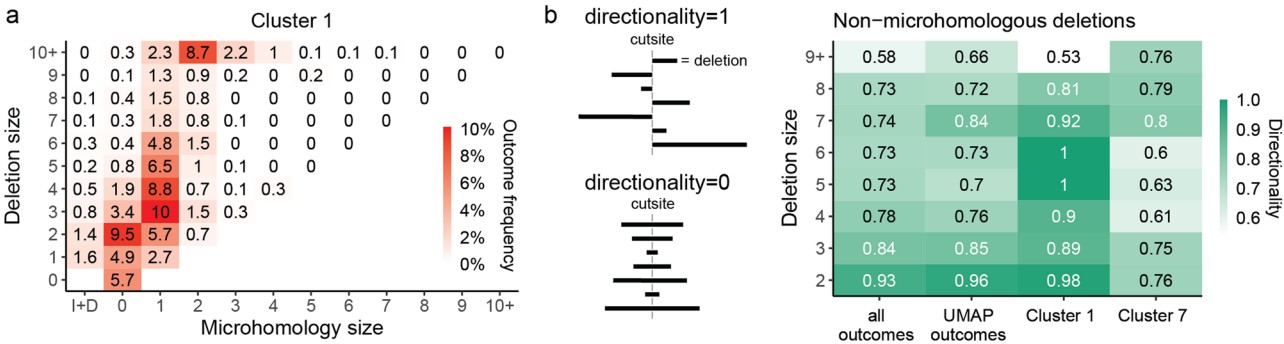

**Fig. 6 | Unidirectional deletions cluster. a** Composition of cluster 1 in terms of microhomology (*x*-axis) and deletion size (*y*-axis). I+D = outcomes that contain both insertions and deletions. Box with 0 bp deletion size and 0 bp microhomology = pure insertions. **b** Directionality of non-homologous deletions. Non-homologous deletions of 1 bp have directionality 1, per definition. Source data are provided as a Source Data file.

profiles was 0.7 (Fig. 7b), with the remaining error primarily driven by targets with no dominant outcomes. Profiles of these targets were also less reproducible experimentally (Supplementary Fig. 7). To consolidate these more variable measurements, we combined the individual outcome forecasts into outcome category groups, which improved performance ($R = 0.77$, Fig. 7b). Finally, we tested the ability of our models to predict the fraction of in-frame mutations as a proxy for predicting the likelihood of generating protein knockouts. This prediction achieved high accuracy ($R = 0.81$, Fig. 7b).

To validate the model performance using measurements generated at an endogenous locus, we used five independently generated mES cell knockout lines (Xrcc5-/-, Polq-/-, Lig4-/-, Poll-/- and Polm-/-)[27]. For each line, we measured Cas9 break repair outcomes at seven different target sites within the *Hprt* locus (Methods). We then calculated Pearson's correlation (*R*) between measured and model-predicted outcome frequencies for matching knockout lines in both the validation dataset and held out data from our original screen. These correlations were similar for individual outcomes ($R = 0.68$ validation vs 0.71 held-out data) and outcome categories ($R = 0.70$ vs 0.81), and slightly better for in-frame fraction ($R = 0.96$ vs 0.81, Supplementary Fig. 8).

## Discussion

We presented the largest assessment of Cas9-induced outcomes in repair deficient contexts to date. We confirmed the crucial roles of *Polq*, *Nbn* and NHEJ genes in modulating the frequency of outcomes generated by DSB repair at a scale, while discovering potentially new dependencies between target sequences and DNA repair. In particular, we elucidated the interaction between flanking nucleotides and DNA repair knockouts in creating 1 bp insertions, leveraged our large dataset to derive clusters of large deletion with different dependencies on *Nbn* and *Polq* and discover a class of non-homologous uni-directional deletions with specific dependency on both *Nbn* and *Xrcc5*.

Single basepair insertions often match the nucleotides flanking the cutsite, which strongly implies they can be templated. The stark enrichment for 1 bp insertions matching the PAM-distal nucleotide in control cells can be due to Cas9 cleavage often resulting in a staggered DSB with 1nt 5′ overhang, a fill-in of which would lead to an insertion of the PAM-distal base[28,29]. A model built by Longo et al. predicted more blunt cuts at sites with a PAM-distal G and staggered cuts at sites with a PAM-proximal G[30]. This implies a lower frequency of insertion at the former type of site and higher at the latter, which is consistent with our data (Fig. 4b). However, in addition to the signal from a flanking G, we observed that a flanking T nucleotide both makes a 1 bp insertion more likely, and inserting a T specifically more likely, which is not predicted by the scission model of Longo et al. Together, this implies that 1 bp insertion outcome is neither independent of the scission profile, nor completely determined by it. We speculate two different scenarios

could lead to an insertion of a templated T. In the first one, a polymerase adds a single T templated in trans on an unresected blunt end. The now staggered end with an extra T itself serves as a template for the blunt end, resulting in two complementary ends that can be ligated. This scenario would imply that unresected ends with a 5′ T (but not an A) serve as good substrates for in trans templated polymerisation. In the second scenario, the break end to be used as a template was partially 3′ resected. This time, two templated additions of a T occur, one after the other, followed by microhomologous annealing of the terminal Ts, fill in and ligation. This could imply that a T (but not an A) is more likely to be resected, more readily exposing a 3′ end for templating. Alternatively, it may be harder to anneal to and ligate after first templated addition (which would result in restoration of wild-type allele), leading to a second round of templated addition and an insertion.

In *Polm* and *Prkdc* deficient cell lines, we observed a specific depletion of 1 bp insertions matching the PAM-proximal nucleotide, without a considerable change in proportion of PAM-distal insertions. We speculate these proximally templated nucleotides are added in trans by a complex of DNA-PKcs and Polμ[31] (Prkdc's and Polm's products) that is bound to the PAM-distal end of the break. This complex may be prevented from binding the PAM-proximal side of the break by Cas9, which remains bound to the PAM-proximal end after the cut, thus explaining the preferential binding and templating. In contrast, we did not observe a similar asymmetry in *Poll* knockouts, implying they are not impeded by Cas9 in their binding[32].

DSB repair is a complex, multi-step, iterative process capable of handling a large diversity of sequence substrates and involving a number of competing as well as co-operating proteins. We took advantage of the large numbers of perturbations and measurements in our screen to cluster the repair outcomes by their response to repair gene knockouts. This approach yielded a number of hypotheses about DSB repair. In particular, it strongly suggested that insertions matching PAM-distal nucleotide may be a result of two different processes, that medium and large deletions follow three different repair patterns, depending primarily on differential *Nbn* and *Polq* involvement and, finally, that uni-directional non-homologous deletions may depend on both *Nbn* and *Xrcc5*, but not on core NHEJ-pathway proteins. Future research may investigate differences between these clusters in more detail. In particular, the question remains whether differences between these clusters correspond to involvement of other specialised repair proteins whose knockouts were not included in this screen, and what particular features of target sequences drive differences in repair.

The assay we used has some limitations. Redundancy of function, especially in the robust NHEJ pathway, is a confounder of all single knockout effects. While consistent modulation of outcomes indicates

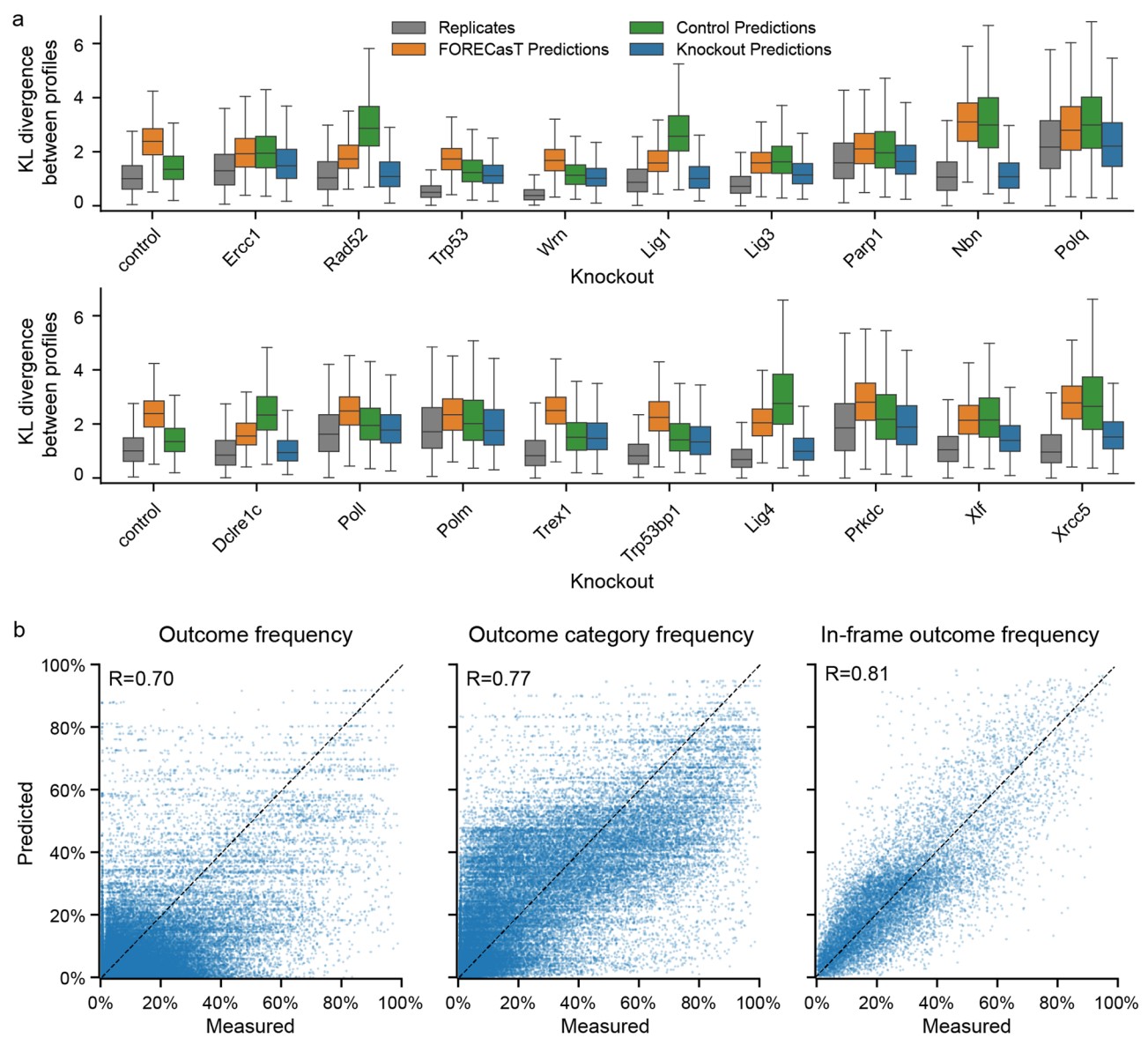

**Fig. 7 | Predictive models of Cas9 mutational outcomes in DNA repair deficient cell lines. a** Distribution of KL divergences between outcome profiles in the same target (y-axis) in each knockout line (x-axis) when comparing measured frequencies to another replicate (grey), to FORECasT predictions (orange), to predictions from the control model (green), and to knockout model predictions (blue). All pairings are from the set of held-out targets (N = 670 targets). Box: median and quartiles; whiskers: 1.5× interquartile range. **b** Measured (x-axis) and predicted (y-axis) frequencies of individual outcomes, outcome categories and in-frame outcomes (columns) in held-out targets (N = 670 targets). R Pearson's R. Source data are provided as a Source Data file.

a causal link, limited signal is not necessarily evidence of a lack of participation in the DSB repair process. For example, both *Poll* and *Polm* in the NHEJ pathway can redundantly perform the required nucleotide synthesis for end processing on certain substrates[33]. A logical next step to further improve understanding is therefore to perform screens in contexts where pairs of genes are perturbed using larger target libraries. In addition, gene-specific sequence features that dictate outcomes could be missing or too rare to detect in aggregate in the library of ~2800 targets. A case in point may be the fact that no particular cluster or outcome category was strongly affected by *Dclre1c* deficiency, even though detailed analysis may imply it specifically affected deletions of size 2 in clusters 1 and 2 (Supplementary Fig. 6). Finally, the assay is limited to measuring outcomes primarily generated by NHEJ and MMEJ, and the effect of HR-associated genes could only be viewed through their modulation of these outcome types. It is known that the rate of HR in mammalian cells is affected by the absence of *Lig4*[34] and it is likely that other genes in the panel also have an effect. Future studies into DSB repair using large scale screens could consider the integration of HR-reporters to improve our understanding of this process.

The activity of repair pathways is the key confounder of all gene editing experiments. The data and models presented in this study shed light on the nature of these interactions for 18 proteins in the cast of DSB repair, but the behaviours and dependencies of the rest remain largely unknown. Similar screens using larger target panels in combinatorial knock-out contexts are needed to cover all relevant repair pathways, and to understand this complex process to predict its results a priori. Accurate predictions combined with options to modulate repair will enable fine control over the outcomes of Cas9-based genome editing.

## Methods

### Library cloning

To generate the library, a 197-mer oligo pool encoding 5760 oligonucleotides was ordered from Twist Bioscience. The library was amplified by PCR using KAPA HiFi HotStart ReadyMix (Roche), 2 ng of template, 0.15 μM subpool forward primers P1 or P2 and 0.15 μM universal reverse primer P3. To reduce the number of polymerase induced mistakes, 10 cycles were used for the PCR. A nested PCR to add GIbson homology ends was done with KAPA polymerase, 0.15 μM primers P4 and P3 over 10 cycles using 2 ng of template DNA. After each PCR, amplicons were purified using Monarch PCR & DNA Cleanup Kit (NEB).

A lentiviral gRNA expression vector lacking the scaffold, pKLV2−U6(BbsI)−PGKpuro−2 A−mCherry−W, was generated by removing the improved gRNA scaffold from pKLV2−U6−gRNA5(BbsI)−PGKpuro−mCherry−W (Addgene 67977; see Allen 2018, with minor changes[7]). The amplicons were cloned into the vector using Gibson Assembly mix reactions (NEBuilder HiFi DNA Assembly Cloning Kit) according to manufacturer's specifications in two or three separate reactions. Gibson reactions were pooled, column-purified and transformed in 4 or 5 electroporations (NEB 10-beta Electrocompetent E. coli C3020K) for a coverage of more than 425×. Bacterial cells were cultured overnight in liquid and plasmid DNA encoding an intermediate library was extracted using QIAGEN Plasmid Maxi Kit (QIAGEN). The vectors were digested with BbsI (NEB).

A 221-mer G-block (IDT) encoding the improved scaffold was amplified using 5 ng of template, KAPA polymerase, 0.1 μM primers P5 and P6 over 25 cycles[7]. The product was column-purified with Monarch PCR & DNA Cleanup Kit (NEB) and digested with BbsI. The intermediate library and G-block were column-purified and ligated (T4 DNA ligase, NEB) in three separate reactions for each subpool. The reactions were combined and digested again with BbsI 37 °C for 30 min to remove any undigested carryover products. The products were column-purified and transformed in either four or five electroporations. Bacterial cells were cultured overnight in liquid and the final libraries were extracted using QIAGEN Plasmid Maxi Kit (QIAGEN). The libraries were quantified and subpools combined in 1:4.76 molar ratio to get the final library containing 5760 gRNAs.

### Cell culture

CAST/BL6 (CB9) mES cells that expressed Cas9 and had a knock-out of a gene in the DNA repair pathway[20], were cultured in M15 media (high-glucose DMEM (Lonza), with 15% FCS (ThermoFisher), 0.1 mM beta-mercaptoethanol, 100 U/ml penicillin and 100 mg/ml streptomycin (Gibco) on SNL-HBP feeder cells. Cells were treated with 10 μg/ml blasticidin for at least 3 days before starting a screen to ensure stable Cas9 expression. The screens were performed without feeder cells in M15 medium supplemented with 1000 U/ml leukaemia inhibitory factor (Merck). Cells were plated on flasks coated with 0.1% gelatin solution (Merck). Medium was changed daily throughout expansion and all experiments. All cell lines were cultured at 37 °C, 5% $CO_2$.

### Lentivirus production and determination of lentiviral titre

Supernatants containing lentiviral particles were produced by transient transfection of 293FT cells using Lipofectamine LTX (Invitrogen). 5.4 μg of a lentiviral plasmid library, 5.4 μg of psPax2 (Addgene 12260), 1.2 μg of pMD2.G (Addgene 12259) and 12 μl of PLUS reagent were added to 3 ml of OPTI-MEM and incubated for 5 min at room temperature. 36 μl of the LTX reagent was then added to the mixture and further incubated for 30 min at room temperature. The transfection complex was added to 80%-confluent 293FT cells in a 10-cm dish containing 10 ml of culture medium. After 48 h viral supernatant was harvested and fresh medium was added. After 24 h the lentiviral supernatant was collected, pooled with the first supernatant, filtered through a 0.45 μm filter and stored at −80 °C.

For lentiviral titration, mES cells were plated into 96-well plates, $5 \times 10^4$ cells per well. 8 μg/ml Polybrene (hexadimethrine bromide, Sigma) was added to each well and the cells were transduced with varying volumes of virus (0 to 20 μl). The cells were then centrifuged at 1000 g for 30 min at room temperature and resuspended in the same media. After three days of cell culture, cells were harvested for FACS analysis and the level of mCherry expression was measured. Data was analysed with Flowjo. Virus titre was estimated and scaled up accordingly for subsequent screens.

### Screening of repair outcomes

mES cell lines were infected aiming for a multiplicity of infection (MOI) of 0.6 to 0.8 and at a coverage 800×. The effective MOI ranged between 0.1–0.6 with a coverage of 100–650× depending on the cell line. For each line, at least two infections were performed and treated as separate biological replicates. Cells were seeded onto 0.1% gelatin coated flasks with a density of $3 \times 10^4$ cells/cm². 24 h after transduction 3 μg/ml of puromycin was added and maintained throughout the screen. Cells were cultured for 14 days after infection. Samples were taken on day 3, 7, 10 and 14 post-infection. Enough cells were passaged and collected to maintain coverage higher than at the time of infection.

### DNA extraction and sequencing library preparation

Upon collection, cells were centrifuged and pellets were stored at −20 °C. For genomic DNA extraction, cell pellets were resuspended into 100 mM Tris-HCl, pH 8.0 (Thermo Scientific), 5 mM EDTA (Invitrogen), 200 mM NaCl (Invitrogen), 0.2% SDS (Promega) and 1 mg/ml Proteinase K (Merck) and incubated at 55 °C overnight. The solution was treated with 10 μg/ml RNase A for 4 h. DNA was extracted by adding one volume of isopropanol followed by spooling, double wash with 70% ethanol and elution in TE buffer overnight. DNA was quantified in triplicate using Quant-iT Broad Range kit (Invitrogen).

For sequencing, the region containing the target surrounded by the context was amplified by PCR using primers P7-P8 with Q5 Hot Start High-Fidelity 2× Master Mix (NEB) with the following conditions: 98 °C for 30 s, 20 cycles of 98 °C for 10 s, 50 °C for 15 s and 72 °C for 20 s, and the final extension 72 °C for 5 min. For each sample, the amount of input gDNA template was adjusted to the screen coverage based on measured MOI and ranged from 35 to 103 μg, aliquoted into 50 μl reactions each containing no more than 5 μg gDNA[7]. The PCR products were pooled in each group and purified using QIAquick PCR Purification Kit (Qiagen). Sequencing adaptors were added by PCR enrichment of 1 ng of the purified amplicons using forward primer P9 and indexing reverse primer P10 with KAPA HiFi HotStart ReadyMix with the following conditions: 98 °C for 30 s, 12–16 cycles of 98 °C for 10 s, 66 °C for 15 s and 72 °C for 20 s, and the final extension 72 °C for 5 min. The PCR products were purified with Agencourt AMPure XP beads. Samples were quantified with Quant-iT 1X dsDNA HS Assay (Invitrogen) and sequenced on Illumina HiSeq2500 or HiSeq4000 by 100-bp paired-end sequencing using Illumina standard primers.

### Data processing of high-throughput knockout screen

Sequencing reads were converted into outcome profiles for each guide using the custom pipeline described in Allen et al. [7], which assigns reads to guides and uses a dynamic programming approach to identify mutations. Guide profiles with less than 100 reads in any knockout, replicate or timepoint were removed from the analysis to ensure adequate coverage. Final mutated read coverages are shown in Supplementary Fig. 1. Outcomes only observed in a single read across all samples were removed. As a result of the filtering, 2838 guides were retained. Two biological replicates for each knockout at each timepoint were combined by pooling together all the reads assigned to the same guide and treating them as one outcome

profile. The three timepoints were combined for each sample in the same fashion as the replicates, as the correlation between them was high.

### Data processing of validation screen at HPRT locus

Sequencing reads were converted into outcome profiles for each guide using the custom pipeline described in Allen et al[7]. Read counts for technical and biological replicates were combined by pooling together all the reads assigned to the same guide and treating them as one outcome profile.

### Clustering

All clustering analyses were performed using outcomes from a set of 2838 targets common to all knockouts. Outcomes not present in the corresponding target in the control cell line or not present in at least 10 knockout lines were removed. UMAP projection of outcome modulation profiles was performed using the umap-learn python package[35] with a min_distance of 0 and a num_neighbors of 50.

### Modelling microhomology dynamics

To model the relationship between outcome frequency ($y$), microhomology size ($s$) and distance between microhomologies ($d$), we fit exponential models of the form $y_s = Ae^{Bd}$ for every size of microhomology from 2 to 15 using the curve_fit function in the scipy python package[36].

### Modelling outcome frequency

A set of possible outcomes and features was generated for each target using the methodology laid out in Allen et al[7]. Generated outcomes included every insertion of up to two nucleotides within 3 nucleotides of the cut site and all deletions of up to 30 nucleotides which span the cut site. 3633 binary features were computed for each outcome, describing their length, location, inserted sequence, involvement of microhomology and nucleotide context, as well as pairwise combinations of these features. These features were paired with the measured frequencies of the generated outcome in our screens. 0.5 reads were added to each outcome for numerical stability. A dataset of generated outcomes, their scaled frequencies in our experiment, and their corresponding features was produced for each target present in each knockout line. These datasets were each randomly split into a training and test set, keeping 10% of the data in each test set. A logistic regression to predict each outcome in a profile was trained by minimising the KL divergence between the predictions of outcomes in a profile and the measured frequencies for all targets in the training set as in ref. 7. Model performance was evaluated by calculating the average KL divergence between measured and predicted profiles in each test set.

### Measuring repair outcomes at endogenous Hprt locus

Previously generated mES cell knockout lines (Xrcc5-/-, Polq-/-, Lig4-/-, Poll-/- and Polm-/-)[27] and wild-type control cells were transfected with plasmid pU6-(BbsI)_CBh-Cas9-T2A-mCherry (a gift from Ralf Kuehn; Addgene plasmid #64324) that co-expressed sgRNAs targeting the Hprt locus (Supplementary Table 2). Cells were transfected in suspension using Lipofectamine 2000 (Invitrogen) using a Lipofectamine:DNA ratio of 2.4:1, incubated for 30 min at at 37 °C and 5% $CO_2$ in round-bottom tubes and subsequently seeded on gelatin-coated plates. Seven days post transfection cells were harvested and used for DNA extraction by lysing pellets in 10 mM Tris-HCL pH 7.5, 10 mM EDTA, 200 mM NaCl, 1% SDS and 0.4 mg/mL Proteinase K and incubating at 55 °C for 16 h. Lysates were neutralised by adding saturated NaCl and DNA was precipitated by adding one volume of isopropanol to the supernatant followed by centrifugation, one wash with 70% ethanol and elution in TE buffer.

For Illumina sequencing, the targeted region was amplified by PCR using target-specific primers (Supplementary Table 2) containing adaptors for the p5 and p7 index primers (5′- GATGTGTA TAAGAGACAG-3′ and 5′-CGTGTGCTCTTCCGATCT-3′ respectively) as previously described[37]. Final PCR products were purified using a 0.8x reaction volume of magnetic AMPure XP beads (Beckman Coulter) and eluted in 20 μL MQ. DNA concentrations were measured using the Quant-iT dsDNA assay kit and the Qubit Fluorometer (Thermo Fisher Scientific) and samples were pooled at equimolar concentrations per Hprt target site. These pools were analysed using a High Sensitivity DNA chip on a Bioanalyzer (Agilent) and equimolar libraries were generated that were sequenced on a NovaSeq6000 (Illumina) by 150-bp paired-end sequencing. Subsequently, SIQ software was used to filter and align NGS-sequence reads to a reference sequence containing the primer sequences and the CRISPR-Cas9 target sites[38].

### Reporting summary

Further information on research design is available in the Nature Portfolio Reporting Summary linked to this article.

## Data availability

Sequencing data: European Nucleotide Archive PRJEB12405. PRJEB39660. PRJEB36814. Processed data: https://figshare.com/s/ce21746028b0b036dea1. Source data are provided with this paper.

## Code availability

Trained models: https://github.com/ananth-pallaseni/FORECasT-repair[39]. Prediction web tool: https://elixir.ut.ee/forecast-repair/.

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

## Acknowledgements

Grant support: A.P., E.M.P., G.G., L.C., and L.P. were supported by Wellcome (220540/Z/20/A). M.K. was supported by NIH grant R01HG003988. I.K., M.M., and H.P. were supported by the Estonian Research Council (grant no. TT11, ELIXIR). J.S. is supported by a Young Investigator Grant from the Dutch Cancer Society (KWF, 2020-1/12925); M.T. is supported by a grant from HollandPTC-Varian (2019020-PROTON-DDR). We would like to acknowledge Allan Bradley and Frances Steward for providing us with the mouse embryonic stem cell DNA damage knock-out library and feeder cells, and for helpful suggestions; Juliane Weller for suggestions on the design of the web tool; and Uku Raudvere for technical guidance.

## Author contributions

Designed project: A.P., L.C., M.K., L.P. Performed experiments: E.M.P., G.G., L.C., N.M.S., and J.S. Analysed and interpreted data: A.P., Ö.S., B.M., M.K., and L.P. Built the website: I.K., M.M., and H.P. Supervised study: M.T., M.K., and L.P. Wrote paper: A.P., M.K., and L.P., with input from all authors.

## Competing interests

B.M. is an employee of Merck Healthcare, Darmstadt, Germany. O.S. is an employee of BioMed X Institute (GmbH), Heidelberg, Germany, which receives research grants from Merck KGaA. L.P. Receives remuneration and stock options from ExpressionEdits. The remaining authors declare no competing interests.
