## [Transparent Peer Review file · Nature Communications]

The interplay of DNA repair context with target sequence predictably biases Cas9-generated mutations

Corresponding Author: Dr Leopold Parts

Version 0:

Reviewer comments:

Reviewer #1

(Remarks to the Author)

This manuscript entitled "The interplay of DNA repair context with target sequence predictably biases Cas9-generated mutations" by Pallaseni et al. using the synthesized Cas9 targets in 21 repair gene knockout mESCs. The current manuscript also determines the outcomes of Indels within the genome and used it to develop a prediction model that allows generating practical ways to induce CRISPR-Cas9 induced mutations.

There are multiple studies trying to illustrate the repair outcome in the context of repair gene knockout, or chromatin status. However, the scope of the gene panel included in such analysis is limited. The current manuscript overcomes this limitation by running a broader analysis on a higher number of cell lines, thereby turning their conclusion more elusive in the context of Cas9 activity. However, we hypothesize that it may have potential some limitations when introduced to broader applications. Overall, the manuscript is carefully written with a logical flow. Most experiments are well designed. Most data are nicely presented, clean and well controlled, and organized in a way to support the main conclusion. However, we advise to improve most of the figure labeling to better correlate with the described findings. And we suggest the authors to develop a validation strategy of selected genomic loci using the repair gene knockout cell lines. The validations in this context are crucial to support the potency of their prediction model.

Major comments

1 Abstracts: Some descriptive points do not match well with the results described in the manuscript. For example;

- 1) It's not soundness to use "Several genes" can have an impact on mutation created. we advise them to stress out better in the abstract.
- 2) The conclusion "Absence of key non-homologous end joining genes Lig4, Xrcc4, and Xlf abolished small insertions and deletions" does not match the data presented. We strongly suggest revising the conclusion presented in this statement.
- 3) The conclusion "Complex alleles of combined insertions and deletions were preferentially generated in the absence of Xrcc6", indicating both insertions and deletions are preferentially generated in the absence of Xrcc6, or it indicates insertions plus deletions group as described in the main text. We strongly suggest a better rephrasing of the concept "insertions and deletion" to fit the group description in the result section.

2 We advise to better clarify the outline of the frequency in each outcome along the manuscript. For example, in Fig. 2a, they present the frequency of the targets however it is unclear whether how they define the highest frequency of 1.0 in this analysis.

3 In Fig.2 the values represented come from an average of multiple targets, and do not reflect in the current form of data presentation possible variability between them. Therefore, we strongly suggest to replace it with a dot plot along with statistical tests and error bar presentation.

4 Fig 2b, we advise in Fig. 2b to replace the analysis with a re-clustering of the repair outcome for the 21. We also suggest dissecting which groups of repair genes are relatively close to each other versus those that are distant. This will better describe the data.

5 In 2021 NAR paper, Gupta group showed repair outcomes in Polq^{-/-} and Ku70^{-/-} cells using Cas9 cutting at mouse Rosa26 locus (Feng et al. NAR, 2021, PMID: 33963863). Consistent with this paper, Ku70^{-/-} decrease the frequency of 1bp-insertion while Polq^{-/-} slightly increased the 1bp-insertion. However, contrary to the long deletion frequency, Gupta's results showed Ku70^{-/-} significantly increase the long deletion frequency. Therefore, we strongly suggest that the authors address these results in an extended manner.

6 Based on the synthetic locus, the authors generated a prediction model, to increase the impact. We strongly suggest that

the authors develop a validation strategy of selected genomic loci using the repair gene knockout cell lines. The validations in this context are crucial to support the potency of their prediction model.

Minor comments

1. We advise to provide the full name of UMAP at the first occurrence in the manuscript.
2. We strongly suggest adding more descriptive details to the legends of panels D, P, N, X in Fig. 4c.
4. In Fig. 4f, There seems to be a typo; Xrcc5 instead of Xrcc4, which we advice to revise it.
5. We strongly advise moving Table. 1 in the manuscript to the data supplementary section. We also advise revising the data presented in this table, especially the relevance between the pathway and the knocked-out gene presented in each option.

Reviewer #2

(Remarks to the Author)

This paper studies the effect of gene knockouts on NHEJ and MMEJ DNA repair outcomes at Cas9-induced double-strand breaks. The authors produce a large dataset that enables quantitative analysis and insights into the roles of various DNA repair genes, and enables training predictive models.

This review primarily considers the computational and machine learning aspects of the work. The predictive models follow proper train/test splitting, and performance metrics such as KL divergence and Pearson correlation are reported in the results and Figure 7. The codebase appears well-written and easy to use, and the website works. Parameters used for the UMAP analyses are reported.

Minor comments:

- I found it interesting that the clustering in the UMAP by outcome category implies that the effect of gene knockouts on repair frequency distributions in a manner is largely independent of sequence context. However, it is not completely independent, as there is still variation within clusters. This suggests to me an opportunity towards building a simple predictive model that can help shed some scientific insight: might it be possible to take a pre-trained FORECasT model, and learn just six parameters, each one a weight for the six different broad categories of DNA repair outcomes considered in the paper, to update/finetune that pre-trained model to be accurate in a specific knockout context? Alternatively, it may be that training new FORECasT models from scratch, at its finer-grained resolution of individual repair outcomes, performs significantly better, which may imply that gene knockouts can differentially impact particular mutations within a shared category. To clarify, these are just musings, not demands.

- Six outcomes per target per cell line seems slightly low, when it is known that these DNA repair outcomes can be highly diverse. Could this be due to uneven or insufficient sequencing depth? To help understand the dataset better for potential future use, I suggest the authors include details in the results or methods section on the average sequencing depth per target per cell line.

- Figure 7b is difficult to read and interpret. Perhaps the authors can consider adding shaded columns on alternating gene knockouts? Also, there are 22 x-axis ticks but 21 knockout genes; the left-most tick is unlabeled and not described in the figure caption.

- On the website, I would suggest renaming the "frequency" axis label to "predicted frequency".

Reviewer #3

(Remarks to the Author)

Reviewer #4

(Remarks to the Author)

Version 1:

Reviewer comments:

Reviewer #1

(Remarks to the Author)

The authors of the manuscript ""The interplay of DNA repair context with target sequence predictably biases Cas9-generated mutations", have nicely addressed most of our concerns that were previously raised. However, since the manuscript encountered cell line mislabeling issues, we strongly advise that the authors provide the cell genotyping results as supplementary data. In regard to the figure representation in the manuscript, we strongly advise the authors to include missing figure captions.

(Remarks on code availability)

Reviewer #2

(Remarks to the Author)

The authors have addressed my review and I have no further comments.

(Remarks on code availability)

Reviewer #3

(Remarks to the Author)

(Remarks on code availability)

Reviewer #4

(Remarks to the Author)

(Remarks on code availability)

We thank the Reviewers for taking the time to provide insightful comments that have prompted us to improve the manuscript, as well as the enthusiasm for the work in general. As a result, we have substantially re-worked the text, and added new findings. In particular, we have:

1. **Re-genotyped all the cell lines** we used in screening, and removed three lines where the measured and expected genotype did not match, including Xrcc6/Ku70.
2. **Validated the machine learning models** on additional data, comparing our predictions against measured editing outcomes from endogenous loci in another set of mouse embryonic stem cells with repair gene knockouts. Prediction accuracy on this data was similar to that obtained on held out test data in the original dataset.
3. Responded to the comments raised, and **improved the text throughout**.

The full response to the raised comments is given below.

Reviewer comment / Our response / Action

Reviewer 1

This manuscript entitled "The interplay of DNA repair context with target sequence predictably biases Cas9-generated mutations" by Pallaseni et al. using the synthesized Cas9 targets in 21 repair gene knockout mESCs. The current manuscript also determines the outcomes of Indels within the genome and used it to develop a prediction model that allows generating practical ways to induce CRISPR-Cas9 induced mutations.

There are multiple studies trying to illustrate the repair outcome in the context of repair gene knockout, or chromatin status. However, the scope of the gene panel included in such analysis is limited. The current manuscript overcomes this limitation by running a broader analysis on a higher number of cell lines, thereby turning their conclusion more elusive in the context of Cas9 activity. However, we hypothesize that it may have potential some limitations when introduced to broader applications.

Overall, the manuscript is carefully written with a logical flow. Most experiments are well designed. Most data are nicely presented, clean and well controlled, and organized in a way to support the main conclusion. However, we advise to improve most of the figure labeling to better correlate with the described findings. And we suggest the authors to develop a validation strategy of selected genomic loci using the repair gene knockout cell lines. The validations in this context are crucial to support the potency of their prediction model.

Major comments

Comment 1.1. Abstracts: Some descriptive points do not match well with the results described in the manuscript. For example;

1. It's not soundness to use "Several genes" can have an impact on mutation created. we advise them to stress out better in the abstract.
2. The conclusion "Absence of key non-homologous end joining genes Lig4, Xrcc4, and Xlf abolished small insertions and deletions" does not match the data presented. We strongly suggest revising the conclusion presented in this statement.
3. The conclusion "Complex alleles of combined insertions and deletions were preferentially generated in the absence of Xrcc6", indicating both insertions and deletions are preferentially generated in the absence of Xrcc6, or it indicates insertions

plus deletions group as described in the main text. We strongly suggest a better rephrasing of the concept “insertions and deletion” to fit the group description in the result section.

Response 1.1. We apologise for the inconsistency in reporting, and thank the Reviewer for pointing this out. After additional quality control of the cell line genotypes (see Response 1.5), we have removed the results for Xrcc4 and Xrcc6, as well as their description, so some of the raised points no longer apply; we have clarified the rest as described below.

Action 1.1. We have revised the text in the Abstract and elsewhere as follows:

1. “Mutagenic outcomes of CRISPR/Cas9-generated double-stranded breaks depend on both the sequence flanking the cut and cellular DNA damage repair. How these two interact has been largely unexplored, limiting our ability to understand and manipulate the outcomes.” [Page 1, line 24-26]
2. We no longer report Xrcc4 results, and report the NHEJ overview as “We found that knockouts of core NHEJ genes Lig4, Xrcc5 or Xlf led to a marked decrease in 1-2 bp insertions” [Page 4, line 123-124] which is well supported by diverse views of the data.
3. We no longer report Xrcc6 results, and thus have removed the sentences in question from the text.

Comment 1.2. We advise to better clarify the outline of the frequency in each outcome along the manuscript. For example, in Fig. 2a, they present the frequency of the targets however it is unclear whether how they define the highest frequency of 1.0 in this analysis.

Response 1.2. We thank the Reviewer for identifying this lack of clarity and have added a description of our frequency metric in the text.

Action 1.2. We added the following text to the first paragraph of the results section (Page 3, lines 95-97: “[We] calculated the frequency of each outcome in each target, defined as the fraction of mutated reads recovered for that target which match that outcome (and is thus bounded between 0 and 100%)”

Comment 1.3. In Fig.2 the values represented come from an average of multiple targets, and do not reflect in the current form of data presentation possible variability between them. Therefore, we strongly suggest to replace it with a dot plot along with statistical tests and error bar presentation.

Response 1.3. We agree that our visualisations highlight a summary across targets, and thus lose out on the details. However, summaries are important to get a global overview of the outcomes. We opt for a multi-pronged approach to 1) show examples of individual outcomes for selected targets in three knockout cell lines in two replicates to gain intuition and appreciate the variability (Figure 2a), followed by 2) summaries of the trends in absolute (Figure 2b) and 3) relative scale (Figure 2c). In addition, we now provide direct comparisons of replicates and targets in the form of a scatter plot (Figure S1c) and outcome type densities (Figure S1d). We note that variation across targets is not trivial to convey meaningfully, as the opportunities for inserting a cut-distal “T” via fill-in, or deleting 6nt via microhomology vary a lot depending on

the target sequence. We therefore show the comparisons of outcome category frequency distributions compared to the control.

Action 1.3. We provide main plots that give intuition about variation across targets and replicates (Figure 2a), as well as consistency across replicates (Figure S1c) and variation of outcome category frequencies across targets in the knockouts and controls (Figure S1d).

Comment 1.4. Fig 2b, we advise in Fig. 2b to replace the analysis with a re-clustering of the repair outcome for the 21. We also suggest dissecting which groups of repair genes are relatively close to each other versus those that are distant. This will better describe the data.

Response 1.4. We thank the Reviewer for the suggestion, and agree that there is useful information in the similarity of the outcomes of individual genes. We have manually ordered the genes to reflect the results of a hierarchical clustering, such that the similar large changes (blue and red entries in the log-fold change heatmap in Figure 2c) are nearby, while constraining the reordering to respect the split of the targeted genes between the various repair pathways.

Action 1.4. We ordered the genes in all plots to reflect both grouping by repair pathway, and the impact on the repair outcomes. Further, we have dissected which repair genes are relatively close to each other versus those that are distant throughout the text, specifically:

- Page 4, lines 123-125 comparing NHEJ genes: “knockouts of core NHEJ genes Lig4, Xrcc5 or Xlf led to a marked decrease in 1-2 bp insertions and 1bp deletion, with concomitant increase in medium and large deletions (3bp+)”
- Page 4, lines 138-144 comparing MMEJ genes: “Polq knockout strongly decreased the frequency of medium deletions with microhomologies compared to control cells (3-9bp; 21% to 7%, Figure 2b), while increasing occurrence of non-homologous large deletions (10bp+; 5% to 17%). [NbN] knockout produced the strongest effect of all tested genes in the our panel, suppressing medium and large deletions (3bp+; 61% to 18%) and resulting in profiles enriched in 1-2 bp insertions and 1bp deletion (25% to 73%)”
- Page 4, lines 148-150 comparing other genes: “The other MMEJ-associated genes in our panel (Lig1, Lig3 and Parp1) and other repair genes (Dclre1c, Wrn, Trex1, Trp53, Trp53bp1, Rad52 and Ercc1) did not substantially affect the major outcome categories”

Comment 1.5. In 2021 NAR paper, Gupta group showed repair outcomes in Polq^{-/-} and Ku70^{-/-} cells using Cas9 cutting at mouse Rosa26 locus (Feng et al. NAR, 2021, PMID: 33963863). Consistent with this paper, Ku70^{-/-} decrease the frequency of 1bp-insertion while Polq^{-/-} slightly increased the 1bp-insertion. However, contrary to the long deletion frequency, Gupta’s results showed Ku70^{-/-} significantly increase the long deletion frequency. Therefore, we strongly suggest that the authors address these results in an extended manner.

Response 1.5. The Reviewer has pointed out an important blindspot in our manuscript, and a relevant reference to add. We have now conducted re-genotyping of our mouse embryonic stem cell knock-out library, using the populations that we used for screening, and discovered that Xrcc6/Ku70, as well as Rnf138 and Bre knockout cells are actually wild-type and that the Xrcc4 clone was mislabelled and is actually an Xrcc5-knockout. All the other clones genotyped

correctly. We have updated the manuscript accordingly by removing additional wild-type clones and renaming Xrcc4 to Xrcc5. In the process, we also thoroughly revised our data filtering and normalization procedures and focused our analysis on the most robust phenotypes.

Action 1.5. We have removed Xrcc6, Rnf138 and Bre clones from the manuscript and renamed Xrcc4 to Xrcc5 to reflect the genotyping results. Given we do not have a Ku70 knockout line, we could not address the results of Feng et al., but now reference their work in the Results section relevant for Polq (page 4, lines 137-139): “Consistent with that role and previous observations (Feng et al, 2021), Polq knockout strongly decreased the frequency of medium deletions with microhomologies compared to control cells (3-9bp; 21% to 7%, Figure 2b).”

Comment 1.6. Based on the synthetic locus, the authors generated a prediction model, to increase the impact. We strongly suggest that the authors develop a validation strategy of selected genomic loci using the repair gene knockout cell lines. The validations in this context are crucial to support the potency of their prediction model.

Response 1.6. We agree with the Reviewer that the strongest validation is on endogenous genomic loci, rather than synthetic ones. We have conducted a validation experiment on endogenous targets without a match in our synthetic library in a set of DNA damage repair deficient mouse embryonic stem cells engineered by a different research group. Predictive performance of our model on the validation data was similar to that obtained on held out data in our screen.

Action 1.6. We have added Supplementary Figure 8 to demonstrate the model performance on externally generated data in knockout mouse cell lines and endogenous loci (copied below as Reviewer Figure 1). We also added the following text to the Results: “To validate the model performance using measurements generated at an endogenous locus, we used five independently generated mouse embryonic stem cell knockout lines (Xrcc5^{-/-}, Polq^{-/-}, Lig4^{-/-}, Poll^{-/-} and Polm^{-/-})²⁷. For each line, we measured Cas9 break repair outcomes at seven different target sites within the Hprt locus (Methods). We then calculated Pearson’s correlation (R) between measured and model-predicted outcome frequencies for matching knockout lines in both the validation dataset and held out data from our original screen. These correlations were similar for individual outcomes (R = 0.68 validation vs 0.71 held-out data) and outcome categories (R = 0.70 vs 0.81), and slightly better for in-frame fraction (R = 0.96 vs 0.81, Supplementary Figure 8).”

Reviewer Figure 1 (Supplementary Figure 8). Measured (x-axis) and predicted (y-axis) frequencies of individual outcomes, outcome categories and in-frame outcomes (columns) in held-out targets in Lig4, Polq, Poll, Polm, Xrcc5 knockouts. Top: original screen (N=670 targets), bottom: endogenous validation screen (N=7 targets). R = Pearson's R.

Minor comments

Comment 1.7. We advise to provide the full name of UMAP at the first occurrence in the manuscript.

Response/Action 1.7. We apologise for omitting the acronym definition, and have added the full name on first its occurrence: “*We embedded the log2 fold changes to 18 gene knockouts of the remaining 18,105 unique outcomes into two dimensions using Universal Manifold Approximation and Projection (UMAP; Figure 3a).*” (page 6, lines 177-178)

Comment 1.8. We strongly suggest adding more descriptive details to the legends of panels D, P, N, X in Fig. 4c.

Response/Action 1.8. We thank the Reviewers for highlighting an important clarification, and have changed panel 4c (now 4a) to better highlight the types of insertions being represented (Reviewer Figure 2 below). The revised panel is given below, emphasising the provenance of the inserted base.

Old figure:

New figure:

Reviewer Figure 2 (Figure 4a). "Most 1bp insertions at the cut site matched the PAM-distal nucleotide in control cells. Dashed line represents Cas9 cutsite between 4th and 3rd basepair upstreams of the PAM sequence. First row represents the case in which basepairs flanking the cutsite are identical to each other, and to the inserted base."

Comment 1.9. In Fig. 4f, There seems to be a typo; Xrcc5 instead of Xrcc4, which we advice to revise it.

Response/Action 1.9. We thank the Reviewer for pointing it out. In the revised manuscript, we have changed the way we analyse the match between 1bp insertion and the cutsite flanking nucleotides and, as a consequence, decided to remove this panel.

Comment 1.10. We strongly advise moving Table 1 in the manuscript to Supplementary Table 1. We also advise revising the data presented in this table, especially the relevance between the pathway and the knocked-out gene presented in each option.

Response 1.10. We agree with the Reviewer that Table 1 is better located in the Supplementary section and have moved it. We describe the relevance between the knockout gene and the pathway in the "Function during DSB repair" column, and indicate this in the table caption.

Action 1.10. We moved Table 1 to Supplementary Table 1. We describe the relevance between the pathway and the knocked-out gene in a column of the table, and added the following text to its caption: "*Pathway assignments for each gene are based on the description of the function presented.*"

Reviewer 2

This paper studies the effect of gene knockouts on NHEJ and MMEJ DNA repair outcomes at Cas9-induced double-strand breaks. The authors produce a large dataset that enables quantitative analysis and insights into the roles of various DNA repair genes, and enables training predictive models.

This review primarily considers the computational and machine learning aspects of the work. The predictive models follow proper train/test splitting, and performance metrics such as KL divergence and Pearson correlation are reported in the results and Figure 7. The codebase appears well-written and easy to use, and the website works. Parameters used for the UMAP analyses are reported.

Minor comments:

Comment 2.1. I found it interesting that the clustering in the UMAP by outcome category implies that the effect of gene knockouts on repair frequency distributions in a manner is largely independent of sequence context. However, it is not completely independent, as there is still variation within clusters. This suggests to me an opportunity towards building a simple predictive model that can help shed some scientific insight: might it be possible to take a pre-trained FORECasT model, and learn just six parameters, each one a weight for the six different broad categories of DNA repair outcomes considered in the paper, to update/finetune that pre-trained model to be accurate in a specific knockout context? Alternatively, it may be that training new FORECasT models from scratch, at its finer-grained resolution of individual repair outcomes, performs significantly better, which may imply that gene knockouts can differentially impact particular mutations within a shared category. To clarify, these are just musings, not demands.

Response 2.1. We agree! On one hand, it is already clear that gene knockouts do differentially impact particular mutations within a category; on the other, the categories in the UMAP are defined by similar impact of mutations, so this is explicitly discouraged. There are many models and datasets available for Cas9 outcome prediction by now. In our particular case, we have 10 outcome categories (I1-2_proximal, D1, etc), but 7 major clusters in the UMAP, and 6 genes (Nbn, Polq, Lig4, Prkdc, Xlf, Xrcc5) that drive the clustering in the UMAP. Therefore, the knockout-specific models of the individual genes for now give the lowest complexity output to describe the outcome space. It will be an interesting machine learning project to learn (ideally shallow and interpretable) transformations that both improve generalisation thanks to data scale, but also elucidate the biological differences between the experiment contexts. We have made both our sequence and count data publicly available to enable this.

Comment 2.2. Six outcomes per target per cell line seems slightly low, when it is known that these DNA repair outcomes can be highly diverse. Could this be due to uneven or insufficient sequencing depth? To help understand the dataset better for potential future use, I suggest the authors include details in the results or methods section on the average sequencing depth per target per cell line.

Response 2.2. We thank the Reviewer for pointing this out. The number of outcomes per target differs between the whole experiment (median 12 per target) and the set of robust outcomes that we used in UMAP analysis (median 6). We have now clarified that in the results section. We have also added a supplementary analysis that shows the number of outcomes per target is consistent across cell lines and controls (Reviewer Figure 3 below). We also show the sequencing coverage per cell line as the Reviewer suggested (Reviewer Figure 4 below), which highlights that we have maintained a minimum of 100 reads per target per cell line. We are happy to clarify these details, and believe the consistency of the observations gives a solid foundation for the analyses in the paper.

Action 2.2. We have amended the number of median outcomes to the unfiltered value of 12, and included Reviewer Figure 3 and 4 as Supplementary Figures 1b and 1a.

Reviewer Figure 3 (Supplementary Figure 1b). Number of outcomes per target (y-axis) for screens in mouse embryonic stem cell lines with different repair gene knockouts (x-axis). Box: median and quartiles; whiskers: 1.5x interquartile range.

Reviewer Figure 4 (Supplementary Figure 1a). Density (x-axis) of number of reads per target (y-axis; logarithmic scale) for screens in mESC lines with different repair gene knockouts (x-axis violins). Horizontal bar: median. Every target had at least 100 reads in every knockout.

Comment 2.3. Figure 7b is difficult to read and interpret. Perhaps the authors can consider adding shaded columns on alternating gene knockouts? Also, there are 22 x-axis ticks but 21 knockout genes; the left-most tick is unlabeled and not described in the figure caption.

Response 2.3. We thank the Reviewer for catching the missing tick label and for suggestions about the clarity of the figure. The missing tick label was the “control” line and we have labelled it as such.

Action 2.4. We added “control” label to the first x-axis tick in Figure 7b (now 7a) and spread the box plots across two rows to give more horizontal space to each gene knockout. The new panel looks as follows:

Reviewer Figure 5 (Figure 7a). Distribution of KL divergences between outcome profiles in the same target (y-axis) in each knockout line (x-axis) when comparing replicates (grey), measured frequencies to FORECasT predictions (orange), measured frequencies to predictions from the control model (green), and measured frequencies to knockout model predictions (blue). Box: median and quartiles; whiskers: 1.5x interquartile range.

Comment 2.4. On the website, I would suggest renaming the "frequency" axis label to "predicted frequency".

Response/Action 2.4. We thank the Reviewer for highlighting the lack of precision in the website axis label, and have amended according to the suggestion.

Reviewer Figure 6. Updated website view.